# Reinforcing Gaps? A Rapid Review of Innovation in Borderline Personality Disorder (BPD) Treatment

**DOI:** 10.3390/brainsci15080827

**Published:** 2025-07-31

**Authors:** Lionel Cailhol, Samuel St-Amour, Marie Désilets, Nadine Larivière, Jillian Mills, Rémy Klein

**Affiliations:** 1Department of Psychiatry and Addictology, University of Montreal, Montreal, QC H3T 1J4, Canada; 2Research Centre of the Montreal University Institute for Mental Health (CRIUSMM), Montreal, QC H1N 3V2, Canada; samuel_st-amour@uqar.ca (S.S.-A.); nadine.lariviere@usherbrooke.ca (N.L.); 3CERVO, Quebec City, QC G1J 2G3, Canada; 4Montreal University Institute for Mental Health, Montreal, QC H1N 3M5, Canada; mdesilets.iusmm@ssss.gouv.qc.ca (M.D.); jillian-louise.mills.cemtl@ssss.gouv.qc.ca (J.M.); 5Department of Health Sciences, University of Quebec at Rimouski, Rimouski, QC G5L 3A1, Canada; 6School of Rehabilitation, University of Sherbrooke, Sherbrooke, QC J1N 3C6, Canada; 7Health and Social Union for Support and Prevention (USSAP), 11300 Limoux, France; rklein@ussap.fr

**Keywords:** personality disorder, psychotherapy, neuromodulation, psychotropic drugs, psychosocial functioning, mortality, suicide, innovation, mortality, comorbidity, brain stimulation, psychotropic

## Abstract

**Background/Objectives**: Borderline Personality Disorder (BPD) involves emotional dysregulation, interpersonal instability and impulsivity. Although treatments have advanced, evaluating the latest innovations remains essential. This rapid review aimed to (1) identify and classify recent therapeutic innovations for BPD, (2) assess their effects on clinical and functional outcomes, and (3) highlight research gaps to inform future priorities. **Methods**: Employing a rapid review design, we searched PubMed/MEDLINE, PsycINFO, and Embase for publications from 1 January 2019 to 28 March 2025. Eligible studies addressed adult or adolescent BPD populations and novel interventions—psychotherapies, pharmacological agents, digital tools, and neuromodulation. Two independent reviewers conducted screening, full-text review, and data extraction using a standardised form. **Results**: Sixty-nine studies—predominantly from Europe and North America—were included. Psychotherapeutic programmes dominated, ranging from entirely novel models to adaptations of established treatments (for example, extended or modified Dialectical Behavior Therapy). Pharmacological research offered fresh insights, particularly into ketamine, while holistic approaches such as adventure therapy and digital interventions also emerged. Most investigations centred on symptom reduction; far fewer examined psychosocial functioning, mortality, or social inclusion. **Conclusions**: Recent innovations show promise in BPD treatment but underserve the needs of mortality and societal-level outcomes. Future research should adopt inclusive, equity-focused agendas that align with patient-centred and recovery-oriented goals, supported by a coordinated, integrated research strategy.

## 1. Introduction

Borderline Personality Disorder (BPD) is a severe and complex psychiatric condition characterised by pervasive instability in emotional regulation, interpersonal relationships, self-image, and impulse control. Individuals with BPD frequently experience intense affective lability, chronic feelings of emptiness, identity disturbance, and recurrent suicidal behaviour or self-injury [1]. Since the seminal study of Linehan [2], substantial progress has been made in the therapeutic management of BPD. A large body of evidence has supported the efficacy of structured psychotherapies [3], and new modalities—both psychological and biological—are continuously emerging. Our understanding of the underlying mechanisms of BPD, its clinical heterogeneity, and the impact on everyday functioning has significantly deepened [4,5]. This growing body of knowledge is now shedding lig on specific therapeutic targets [4,6,7]—mortality, symptoms, psychosocial functioning, and social inclusion—which will be explored in the following section, alongside emerging personalised interventions.

Among all treatment priorities, life expectancy stands out as the most fundamental. While death is an inherent part of life, the stark inequalities in life expectancy between individuals with BPD and the general population are both striking and unacceptable [8,9,10]. Reducing this gap is imperative because, without life, no other therapeutic goal retains its meaning. Historically, efforts to extend life expectancy have focused primarily on lowering suicide, which remains the most prominent cause of excess mortality in individuals with BPD compared to the general population [9,11]. Consequently, suicide prevention has become a central focus of numerous therapeutic interventions. While several treatments have demonstrated effects on suicidal ideation, self-injurious behaviour, and suicide attempts [3,12], it is important to recognise that no intervention to date has shown a definitive impact on reducing suicide mortality itself. Moreover, a critical but often overlooked reality is that most individuals with BPD ultimately die from physical health conditions—many of which are both preventable and more prevalent in this population than in the general population (e.g., cardiovascular disease, substance-related disorders, metabolic syndrome) [8,9].

Beyond its potential to mitigate mortality risks associated with both BPD and comorbid conditions, symptom reduction also addresses the core reasons individuals seek psychiatric care, making it a critical therapeutic priority. Treating BPD involves alleviating the emotional distress and functional impairments associated with core BPD symptoms, along with those stemming from common comorbidities. Over the past decades, most clinical studies have focused on reducing BPD symptomatology as the primary treatment target [3,13]. However, there is a growing recognition that comorbid conditions—such as depression, anxiety, PTSD, substance use disorders, ADHD and eating disorders—are not only highly prevalent in individuals with BPD [14], but also significantly influence clinical outcomes, including relapse [15], quality of life [16], and suicide risk [17,18]. Acknowledging the central role of comorbidities underscores the importance of considering their reduction and management as essential therapeutic outcomes.

The third level of treatment targets broader, person-centred outcomes such as quality of life, sense of purpose, functional recovery, and social integration. For many individuals with BPD, the pursuit of a fulfilling life—encompassing gratifying relationships, stable employment, meaningful activities and projects, and a coherent life narrative—remains a significant challenge [19] and a goal [20]. While symptomatic remission is often achievable, complete recovery, defined by both clinical and functional remission, is far less common [21]. Despite their importance, these outcomes have traditionally been treated as secondary endpoints in clinical research, often overshadowed by symptom reduction and safety measures. This raises important questions about treatment development and evaluation priorities: Are we aiming for mere symptom control or for the possibility of functioning optimally in everyday activities and thriving in a meaningful life? Addressing this gap calls for a shift toward long-term, holistic goals that reflect patients lived experiences and aspirations.

A fourth level of treatment targets operates at the interpersonal and societal level, reflecting the bidirectional relationship between individuals with BPD and their social environment. It is well established that early relational trauma—particularly during childhood and adolescence—plays a central role in the development of BPD [22]. Later in life, social stigma, especially within healthcare systems, can further exacerbate suffering and reinforce exclusion [23]. Conversely, BPD significantly affects others. As a disorder fundamentally rooted in interpersonal dysfunction, BPD can place considerable strain on families [24], partners, children [25], friends, and professional networks. At a broader level, the disorder poses challenges to healthcare systems and society, contributing to high rates of service use, emotional burden on caregivers, and workplace difficulties [26]. Acknowledging and addressing this social dimension is essential. Therapeutic innovations must not only aim to support the individual but also repair and strengthen social bonds, reduce stigma, and consider systemic impacts, ultimately fostering healthier relational ecosystems around the person with BPD.

While research continues to provide growing evidence on how best to treat individuals with BPD, efforts often remain fragmented. Each study pursues its own path, focusing on specific aspects of the disorder without contributing to a unified understanding. Despite the clinical and societal burden of BPD, research funding remains limited [27], and there is currently no coordinated strategy or collective roadmap to address the four central domains of treatment outcomes: premature mortality, symptom reduction, psychosocial functioning, and societal impact. This fragmentation also impacts clinical care [28]. People with BPD and their families consistently express a desire for high-quality, professional healthcare that is rooted in respect, compassion, and meaningful therapeutic relationships [29]. This rapid review examines whether current therapeutic innovations truly reflect and respond to the full complexity of BPD.

The objectives of this rapid review were threefold. First, to systematically identify and classify recent therapeutic innovations for BPD, including novel psychotherapies, pharmacological treatments, digital tools, and neuromodulation techniques, published between 2019 and 2025. Second, to evaluate how these interventions address key clinical and functional outcome domains, such as mortality (e.g., suicide prevention and reduction in physical health-related mortality), symptomatology (targeting core BPD symptoms and common comorbidities), psychosocial functioning (including recovery, quality of life, and global remission), and societal impact (such as stigma reduction, interpersonal relationships, and occupational functioning). Third, this review aimed to identify persisting research gaps and misalignments between current innovation trends and pressing therapeutic needs, to inform future research priorities and promote a more socially responsive and outcome-relevant research agenda.

## 2. Materials and Methods

### 2.1. Rapid Review Methodology

This study employed a rapid review approach, defined as a form of knowledge synthesis that accelerates the systematic review process by streamlining or omitting specific steps to deliver evidence in a resource- and time-efficient manner. As outlined by the Cochrane Rapid Reviews Methods Group [30], rapid reviews are beneficial when timely evidence is needed for urgent or high-priority decision-making. In this specific case, our goal was to provide our research community with timely insights so they could identify any potential gaps in the protocol and address them as needed. Methodological adaptations may include narrowing eligibility criteria, limiting the number of databases searched, simplifying screening and data extraction processes, and using single-reviewer approaches with verification.

### 2.2. Eligibility Criteria

Inclusion Criteria
○Population: Adults or adolescents diagnosed with BPD (DSM-5, ICD-10/11, or equivalent standardised diagnostic criteria).○Intervention: Any intervention deemed innovative (mutual agreement at selection based on treatment development knowledge), including but not limited to novel pharmacological agents, new psychotherapeutic modalities, digital interventions, neuromodulation techniques, or microbiome-based approaches.○Publications dated between 1 January 2019 and 28 March 2025.○Original empirical studies of the following designs: randomised controlled trials (RCTs), quasi-experimental studies, feasibility or pilot studies, and prospective or retrospective observational cohorts.○Language: Full-text articles published in English or French.
Exclusion Criteria
○Mixed-diagnosis populations without separate BPD results.○Paediatric samples (<12 years).○Non–peer-reviewed material (e.g., protocols, abstracts, editorials, letters, opinion pieces).○Reviews without original data.○Studies of other personality disorders without separate BPD analyses.


### 2.3. Search Strategy

An expert in mental health literature searching (MD) conducted comprehensive electronic searches across three databases: PubMed/MEDLINE, PsycINFO, and Embase. The goal was to identify all studies of innovative interventions for BPD published between 2019 and March 2025.

Keyword Development (Appendix A)
-Borderline personality disorder: Descriptors: “Borderline Personality Disorder” [Mesh] or keywords (titles abstracts): Borderline personality(ies), Borderline state(s)-Studies: Descriptors: “Clinical Study” [Publication Type] or keywords (titles abstracts): Clinical trial(s), Clinical study(ies), Randomised-control trial(s), Observational study(ies)
Search Documentation and Management

All search results were imported into Covidence, where duplicate records across databases were automatically identified and eliminated. A master list of unique citations was maintained, with each entry annotated based on its database of origin.

### 2.4. Study Selection

Two reviewers (SS and LC) independently screened all titles and abstracts for potential inclusion. Each reviewer applied the predefined eligibility criteria. Conflicts at the title/abstract stage were resolved by consensus. Studies that passed title/abstract screening underwent a full-text review by SS and LC. During full-text review, reasons for exclusion (e.g., incorrect population, out-of-scope interventions such as non-innovative treatments, non-empirical studies) were documented in a standardised exclusion log. The overall selection process is illustrated in a PRISMA flow diagram (Figure 1).

### 2.5. Data Extraction

A standardised data extraction form was developed in Covidence. A first extraction was performed using ChatGPT 04-mini (prompt in Appendix A), and then manually verified (LC).

## 3. Results

### 3.1. Sample

The 69 included studies were conducted primarily in Europe. Participant ages ranged from 15.7 to 68 years, with the samples predominantly composed of women (see Appendix A and Figure 2).

Study designs differed significantly, with RCTs comprising 52.2% of the studies and case reports accounting for 17.4%. Most studies were pilot or feasibility studies. Twelve publications (17.4%) included vulnerable populations, primarily adolescents (n = 6), but also gender minorities (n = 2), individuals in the perinatal period (n = 1), age over 65 (n = 1), hospitalised patients (n = 1), and those with treatment-resistant conditions (n = 1).

### 3.2. Cataloguing Therapeutic Innovations

The studied interventions were mainly psychotherapeutic programmes (n = 40; 56.3%), followed by psychotropic medications (n = 13; 18.3%), neuromodulation techniques (n = 6; 8.5%), digital tools (n = 6), and other psychosocial methods (n = 6) (Table 1). Some studies examined interactions between different treatment methods, such as combining neuromodulation with psychotherapy [31]. The length of interventions varied greatly, from a single session to two years, with a median duration of 13 weeks.

Regarding psychotherapeutic innovations, research teams have explored various strategies. These include extending the duration of standard treatments, such as an enhanced form of DBT, in cases of long-lasting BPD symptoms, with a focus on improving functioning rather than just alleviating symptoms [32]. Additionally, they have addressed frequent comorbidities of BPD using established interventions that show effectiveness in treating both the comorbidity and BPD symptomatology (e.g., Eye Movement Desensitization and Reprocessing (EMDR) for PTSD [33,34], Imagery Rehearsal Therapy (IRT) for nightmares [35,36]). Other strategies involve validating specific techniques, such as memory reconsolidation [37] and mental imagery [38], and implementing more novel approaches, like adventure therapy [39,40]. Notably, this last intervention targeted both symptomatic and metabolic dimensions of BPD, representing one of the few holistic approaches identified in the field. Other perspectives are also emerging—for example, couple therapy [41] offers the opportunity to address not only a core feature of BPD, namely relational instability, within an ecological framework, but also the often significant relational consequences experienced by close others. Some additional therapeutic approaches incorporate bodily experiences and emotional awareness—examples include yoga-based interventions [42] and dance movement therapy [43]. Other innovations explore new directions, such as Metacognitive Interpersonal Therapy (MIT) [44,45], and extended DBT protocols for non-responsive cases [46]. Finally, we highlight the integration of DBT within Assertive Community Treatment (ACT) frameworks, which are typically designed for individuals with more severe presentations [47].

In terms of psychopharmacology, several drugs were investigated, including cariprazine (n = 1) [48], brexpiprazole (n = 2) [49,50], clozapine (n = 1) [51], buprenorphine/naloxone (n = 1) [52], memantine (n = 1) [53], and (es)ketamine (n = 5) [54,55,56,57,58]. (Es)ketamine was administered either for core BPD symptoms or for comorbid conditions such as major depressive disorder [56,57] or bipolar depression [55]. Findings for this drug were mixed; however, several studies reported encouraging results regarding reductions in depressive symptoms and suicidal ideation. Notably, one study identified a potential increase in impulsivity as an adverse effect [55]. The clozapine study was unable to recruit enough participants due to the COVID-19 pandemic, which prevented the drawing of any definitive conclusions [51]. The cariprazine report was a single-case study that showed promising effects on core BPD symptoms [48]. Based on a case series, brexpiprazole may hold potential for addressing aggression, suicidality, and substance use in individuals with BPD [49].

In the field of neuromodulation, recent repetitive Transcranial Magnetic Stimulation (rTMS) studies have explored various stimulation sites, including the dorsomedial prefrontal cortex [59,60] and the right orbitofrontal cortex [61], yielding promising preliminary results. One team researched the combination of theta-burst stimulation, a more intensive rTMS treatment, with DBT but found limited synergistic effects, likely due to the already strong efficacy of DBT alone [31]. A notable advancement in this field is the increasing utilisation of transcranial direct current stimulation (tDCS), a considerably more affordable technique than rTMS with minimal side effects. Preliminary findings indicate that anodal stimulation of the left dorsolateral prefrontal cortex may enhance emotional regulation [62], while stimulation of the right dorsolateral prefrontal cortex may decrease impulsivity [63]. Significantly, tDCS modulates brain activity during stimulation, providing promising opportunities for synergy with specific tasks, psychotherapeutic interventions, or cognitive remediation exercises.

On the digital innovation front, two main approaches can be distinguished: fully self-directed interventions, in which patients follow an online therapeutic programme independently [64], and digital tools designed to enhance traditional psychotherapeutic experiences, such as DBT [65]. One study based on the first approach, grounded in the theoretical model of schema therapy, offers promising perspectives for population-level interventions [64]. Preliminary results suggest a small effect size, which appear to diminish over time.

Other psychosocial studies encompass innovations that are challenging to categorise within the previous domains but are likely to hold significant value, such as the Family Connections programme [66] and the implementation of a national suicide hotline specifically for individuals with BPD [67]. Cognitive training, including emotional working memory interventions [68], provides a novel perspective to the field of cognitive remediation. In the perinatal period, new evidence [69] opens the field for supporting mothers with brief emotional interventions inspired by DBT. Finally, one injection of botulinum toxin has been explored in recent trials [70,71], although it does not appear to be superior to acupuncture.

### 3.3. Assessing Outcome Domains

Globally, the outcomes reported across studies indicate a hierarchical focus—from symptom reduction to broader societal impact—encompassing recovery and mortality-related domains (Table 2).

When mortality is assessed, which is the case in a significant portion of studies (n = 31; 44.9%), it primarily focuses on suicide risk. However, most often, this is limited to proxy indicators (such as suicidal ideation or self-threatening behaviours) rather than direct indicators (suicide rate). While no studies directly measured all-cause physical mortality, some interventions did target metabolic or cardiovascular risk factors [39], reflecting an emerging concern for physical health outcomes in this population.

Symptomatology is by far the most frequently assessed domain (n = 57; 82.8%), with BPD-specific symptoms often serving as the primary outcome. This predominant focus on clinical symptom reduction is typically accompanied by measures of psychosocial functioning or quality of life, which were included in 34 studies (49.3%), although almost exclusively as secondary outcomes.

Finally, societal impact—including factors such as healthcare costs, interpersonal relationships, job tenure, and experiences of stigma or discrimination—was addressed in only 10 studies (14.5%). 

## 4. Discussion

To our knowledge, this is the first rapid review specifically focused on new treatment avenues in BPD research in the past 5 years. Our findings indicate that studies on innovative interventions have been published at a regular rate. These studies predominantly involve young women living in the western world and primarily explore psychotherapeutic strategies aimed at reducing BPD symptoms or associated comorbidities.

While these interventions offer promising perspectives and foster hope for improved care, they appear only partially aligned with both the subjective and objective needs identified in the field. From the perspective of individuals living with BPD, key unmet needs include reducing discrimination, particularly in general medical settings, addressing geographic disparities in access (e.g., in rural areas), and improving support for gender and community inclusion [72]. On the objective side, critical issues such as elevated mortality rates [8,9,10] and marked deficits in psychosocial functioning [19,73] remain largely unaddressed as primary targets in current research. However, this misalignment has previously been highlighted by several researchers, who have emphasised the need for specific programmes that are more closely aligned with recovery-oriented goals [74].

These gaps reflect not only a misalignment of research priorities but also point to a broader systemic problem in the way innovation is conceived, funded, and implemented in the field of BPD. Structural patterns in research funding allocation may reinforce this systematic bias [27]. This systemic issue may be driven by entrenched social conceptions of the disorder, implicit biases about who is considered “worthy” of care, and the institutional interests or limitations of research teams. Addressing this disconnection will require a deliberate shift toward more inclusive, needs-based, and equity-oriented research agendas.

Highlighting these gaps does not diminish the significant potential of current innovations, especially when viewed through a staged-care perspective tailored to the heterogeneity of the BPD population. For example, digital interventions and national hotlines may offer valuable tools to ensure broad coverage of symptomatic and suicidality-related needs at the population level. Brief psychotherapies targeting key comorbidities, such as PTSD, or core symptoms like emotional dysregulation, along with low-cost and well-tolerated biological interventions like tDCS, could serve as accessible options for individuals with moderate symptom severity. Simultaneously, emerging biological treatments such as ketamine and more precisely targeted rTMS protocols may present promising avenues for individuals with more severe or treatment-resistant cases. Finally, interventions focused on employment and family functioning serve as important adjuncts to be integrated in current clinical settings, contributing to a more comprehensive and recovery-oriented approach to care.

The mapping of these therapeutic innovations provides, in filigree, an overview of current conceptualizations of BPD. On the one hand, interventions that integrate multiple therapeutic modalities remain rare, reflecting a persistent preference for comparative approaches over personalised or modular frameworks. Although dimensional diagnostic models are increasingly discussed in the literature [75], most treatment strategies continue to rely on categorical frameworks—consistent with the scope of this review—while the integration of dimensionality into therapeutic design remains limited. From a psychological perspective, no breakthrough has emerged in recent years regarding the conceptual foundations of BPD. Core domains such as relational difficulties, identity disturbance, and emotional dysregulation remain central to both understanding and intervention. The field continues to evolve around these axes without proposing a unified conceptual shift. In the pharmacological domain, dopamine antagonists remain the most frequently studied agents, despite limited support from meta-analyses [13]. Yet, emerging interest in NMDA receptor modulation—particularly via agents such as memantine and ketamine—suggests a shift toward new neurobiological models [76]. Neuromodulation not only opens novel therapeutic avenues but also contributes to the identification of neural circuits involved in emotional regulation [62], social rejection [77], and impulsivity [63]. Digital interventions, while promising in terms of scalability and cost-effectiveness [65], do not introduce new conceptual paradigms. Instead, they primarily rely on established psychotherapeutic frameworks. Several emerging approaches, however, explore bodily dimensions of BPD, including facial expressivity modulation through botulinum toxin [70,71] or embodied therapies [43], potentially enriching our understanding of mind–body interactions in the disorder. Finally, the neuropsychological dimension of BPD is gaining renewed attention. While classic cognitive remediation approaches are being re-evaluated, cognitive training interventions such as those proposed by Krautz [68] offer a transdiagnostic and developmentally sensitive perspective. Promising links are emerging with family- and child-focused approaches, as well as with couple-based interventions. Thus, while the last few decades have addressed the challenge of improving individual skills of people living with BPD (e.g., mentalization, emotional regulation), social functioning and roles are emerging as increasingly desirable and plausible targets for intervention. This shift underscores the importance of considering social roles in the therapeutic process to enhance overall functioning and integration.

This review has several limitations that may collectively underestimate the innovations published in recent years. Due to the selection of publications focused on specific measures of symptoms or functioning, we may not fully capture emerging perspectives from early-stage research, such as those related to lifestyle medicine [78]. Additionally, defining and qualifying “innovation” posed challenges during the selection process. While we attempted to reach a consensus, it was inherently influenced by the authors’ backgrounds. This subjectivity might have led to decreased sensitivity, causing us to miss some opportunities or overemphasise specific dimensions. For example, one might question whether extending the duration of DBT truly constitutes an innovation [46]. Also, some approaches, such as the use of memantine [53], do not represent recent developments, yet they remain outside clinical guidelines and meta-analytic summaries, raising questions about their positioning within the innovation landscape. We should be aware that innovations in other fields, such as substance use disorder treatment, suicide prevention, or emotional regulation, could significantly impact the outcomes for patients with BPD. However, these impacts might not be captured by the narrow methodology of a rapid review. Finally, readers should be aware that we concluded our literature review in March 2025. However, research in this field is ongoing, and as a result, more recent developments, such as those related to peer support [79], are not covered in this paper.

## 5. Conclusions

This rapid review highlights a range of recent promising approaches in the treatment of BPD, all of which require replication and further validation to establish their effectiveness. Nonetheless, these developments offer valuable insight into the evolving direction of clinical research, and clinicians are encouraged to stay informed to prepare for future implementation.

From a research perspective, we urge professional associations to promote greater alignment in research priorities and to actively involve people with lived experience in setting these priorities. This includes addressing persistent gaps in population representation, such as gender and age disparities in recruitment, as well as fostering research in underrepresented regions, particularly in Africa and South America. We also advocate for increased support of breakthrough innovations tailored to varying levels of symptom severity and mortality risk (especially physical health), as well as those focused on recovery processes. These efforts are essential to building a more equitable, diverse, and stage-sensitive evidence base.

Looking ahead, we recommend establishing dedicated task forces that include individuals with lived experience. Such groups could provide ongoing oversight, ensure vigilance regarding the ethical and social implications of research, and offer guidance to align future studies with principles of social responsibility and meaningful, patient-centred outcomes.

## Figures and Tables

**Figure 1 brainsci-15-00827-f001:**
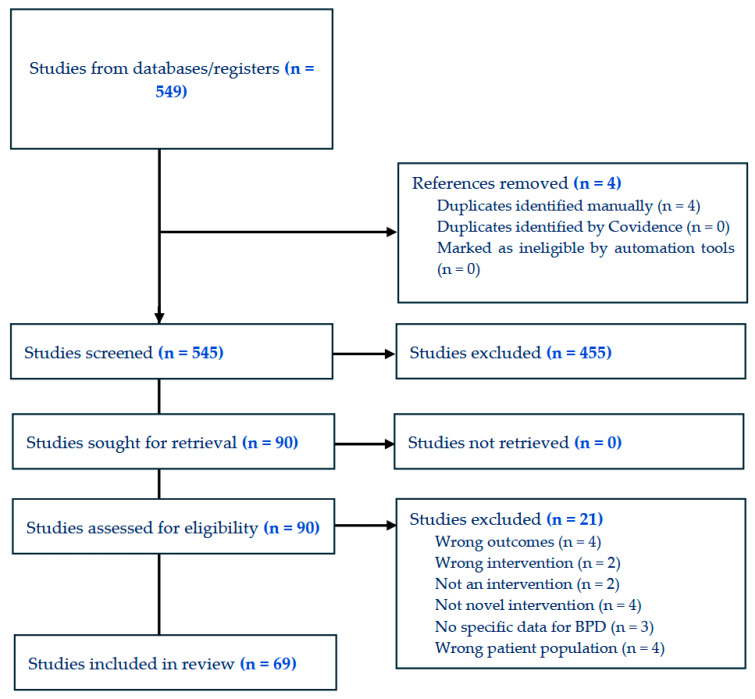
PRISMA flow-chart.

**Figure 2 brainsci-15-00827-f002:**
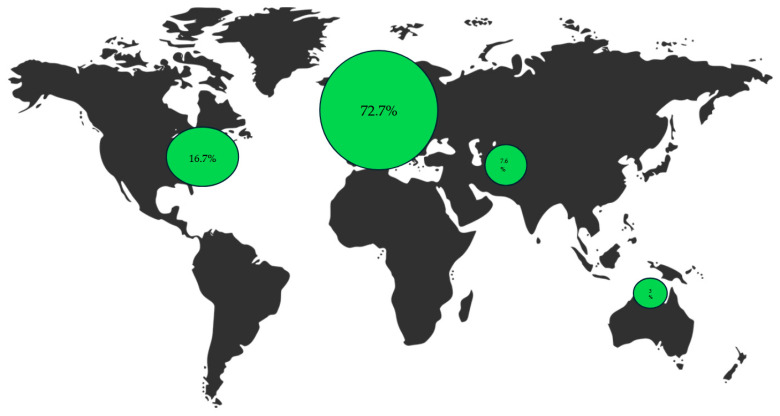
Proportion (%) of studies included in this rapid review conducted by country (a single study may be conducted in multiple countries). Europe (72.7%), North America (16.7%), Asia (7.6%), and Oceania (3%).

**Table 1 brainsci-15-00827-t001:** Overview of 69 Included Studies on Therapeutic Innovations in Borderline Personality Disorder (Since 2019).

Study ID	Study Design	Mean Age (SD) Intervention	Sex (Female %) Intervention	Psychotropic Drug (Y/N) Intervention	Neuromodulation (Y/N) Intervention	Psychotherapy (Y/N) Intervention	Digital Tools (Y/N) Intervention	Other: Specify Intervention	Duration (Weeks/Months) Intervention
Tinlin-Dixon 2024	Case report	68	100	N	N	Y	N	N	16 weeks + 4 follow-up sessions
Hansen 2022	Case report	26	100	Y	N	N	N	N	15 months
Mohajerin 2025	Randomised controlled trial	15.91 (0.98)	65.2	N	N	Y	N	N	12 weekly sessions
Reinsberg 2023	Non-randomised experimental study	24	86.7	N	Y	N	N	N	4 weeks
Sosic-Vasic 2024	Randomised controlled trial	25.58 (5.7)	100	N	N	Y	N	N	2 weeks
Kleindienst 2021	Randomised controlled trial			N	N	Y	N	N	12 months high-frequency + 3-month booster
Gabarda-Blasco 2024	Non-randomised experimental study	39.5	65.4	N	N	Y	N	N	14 weeks
Galuszko-Wegielnik 2023		26	100	Y	N	N	N	N	4 weeks
Molavi 2020	Randomised controlled trial	30.69 (5.01)	50	N	Y	N	N	N	10 days
Back 2022	Randomised controlled trial	29 (7.9)	100	Y	N	N	N	N	one session
Danayan 2023	Cohort study	38.8 (14.6)	58	Y	N	N	N	N	2 weeks
Mohajerin 2024	Randomised controlled trial	27.16 (3.73)	56.6	N	N	Y	N	N	UP: ~42 months
Rothman 2024	Randomised controlled trial			Y	N	N	N	N	11 weeks (randomised phase)
Mendo-Cullell 2021	Non-randomised experimental study	41.5	50	N	N	Y	N	N	14 weeks
Fineberg 2023	Randomised controlled trial	32.1 (10.4)	83.3	Y	N	N	N	N	Single infusion (40 min)
Fitzpatrick 2025	Cohort study	31.31 (7.45)	75	N	N	Y	N	N	12 weeks
Moran 2024		29 (4)	100	N	N	Y	N	N	Two 1 h preparatory + 12 weekly 2 h group sessions (~14 weeks)
Steuwe 2021	Randomised controlled trial	30.82 (8.34)	100	N	N	Y	N	N	10 weeks
Laursen 2021	Randomised controlled trial			N	N	N	Y	N	40 weeks at 2 sites; 12 months at 3 sites
Austin 2020	Other: Mixed: qualitative and quantitative design	28.9 (6.7)	95	N	N	Y	Y	N	20.3 ± 6.3 weeks during DBT programme
Guillén 2022	Non-randomised experimental study	55.57 (8.9)	61.5	N	N	N	N	Family	8 weeks
Rossi 2023	Randomised controlled trial	28.1 (7.4)	86.5	N	N	Y	N	N	12 months
Salvatore 2021	Case report	mid-30s	100	N	N	Y	N	N	18 months
Vaz 2020	Case report	20	100	N	N	Y	N	N	12
Herpertz 2020	Randomised controlled trial	29.8 (9.5)	66.7	N	N	Y	N	N	6 weeks
Rogg 2023	Case report	22	100	Y	N	N	N	N	9 months
Vanicek 2022	Case report	20	100	Y	Y	Y	Y	Y	2 weeks
Bozzatello 2020	Randomised controlled trial		68.4	N	N	Y	N	N	10 months
Bo 2022	Case report	16	100	N	N	Y	N	N	14 months
Calderón-Moctezuma 2020	Randomised controlled trial	24 (6.29)	71.43	N	Y	N	N	N	3 weeks
Buronfosse 2023	Randomised controlled trial			N	N	N	N	Hotline	12 months
Sayk 2025	Non-randomised experimental study	29.9 (9.61)	100	N	N	Y	N	N	8 weeks (8 sessions)
Hurtado-Santiago 2022	Randomised controlled trial	21.10 (4.3)	90	N	N	Y	N	N	10 weekly group sessions (90 min) + 3 follow-up group sessions
Bozzatello 2021	Randomised controlled trial	NR	NR	N	N	Y	N	N	10 months
Bozzatello 2023	Randomised controlled trial	32.89 (10.64)	71.4	N	N	Y	N	N	20 weeks
Dunand 2025	Other: A multiple case study of clients’ experiences	32	83.3	N	N	Y	N	N	5–20 months of IPS support (per client)
Guillén 2024	Randomised controlled trial	56.89 (10.5)	64.9	N	N	N	N	Family	12 weeks
Harty 2024	Case report	NR	100	N	N	Y	N	N	2 months
Dwyer 2025	Randomised controlled trial			Y	N	N	N	N	12 weeks
Kujovic 2024	Randomised controlled trial	24.8 (5.9)	88.2	N	Y	Y	N	N	4 weeks (20 sessions)
Schulze 2024	Randomised controlled trial	28.75 (5.93)	100	N	N	N	N	Botulinium	Single administration; follow-up 4 weeks
Arntz 2022	Randomised controlled trial			N	N	Y	N	N	24 months
Alavi 2021	Non-randomised experimental study		80.8	N	N	Y	Y		15 weeks
Hilden 2021	Randomised controlled trial	31 (8.8)	92	N	N	Y	N	N	20 weeks (20 sessions over 5 months)
Krause-Utz 2020	Randomised controlled trial			N	N	N	N	Cognitive training	28 days (min 16, max 20 training days)
Schmeck 2023	Non-randomised experimental study	16.22 (1.57)	91	N	N	Y	N	N	6–8 months
Klein 2021	Randomised controlled trial			N	N	N	Y	N	12 months
Quattrini 2025	Randomised controlled trial	28 (8)	91	N	N	Y	N	N	12 months
Chanen 2022	Randomised controlled trial			N	N	Y	N	N	Up to 16 sessions (weekly) or until 6-week nonattendance
Kehr 2024	Case report	40	100	N	N	Y	N	N	8 sessions over 2 months
Feffer 2022	Randomised controlled trial	33.9 (9.8)	100	N	Y	N	N	N	
Riegler 2023	Non-randomised experimental study	34.3 (10.05)	73.2	N	N	Y	N	N	8 weeks
Assmann 2025	Randomised controlled trial	29	90	N	N	N	Y	N	12 months
Schindler 2024	Randomised controlled trial	31.7 (10.2)	78	N	N	Y	N	N	12 months
Mohammadsadeghi 2023	Randomised controlled trial	26.85 (8.63)	45	Y	N	N	N	N	8 weeks
Salamin 2021	Cohort study	34.5 (10.4)	85.5	N	N	Y	N	N	2 years (repeated one-year programme)
Juul 2022	Case report	28	100	N	N	Y	N	N	20 weeks
Wollmer 2022	Randomised controlled trial	30.44 (5.80)	100	N	N	N	N	Botulinum toxin A	Single treatment at baseline; follow-up assessments to 16 weeks
Crawford 2022	Randomised controlled trial	28 (7.54)	73	Y	N	N	N	N	Up to 6 months of treatment (follow-up period)
Lisoni 2020		38 (10.9)	53.3	N	Y	N	N	N	3 weeks (15 sessions)
Hafkemeijer 2023	Case report	Non reported	100	N	N	N	N	N	4 days
Grant 2020	Case report	42	0	Y	N	N	N	N	7 months
Francis 2024	Case series	30.3	100	Y	N	N	N	N	4 weeks
Hood 2024	Randomised controlled trial			N	N	Y	N	N	18 weeks (18 sessions)
Sauer-Zavala 2023	Randomised controlled trial	33.71 (13.96)	84	N	N	Y	N	N	18 sessions (within 7-month window)
Soler 2022	Qualitative research	40.3 (6.1)	95	N	N	Y	N	N	12 weeks
Other: Non-concurrent multiple baseline single-subject design		30.6 (12.4)	83	N	N	Y	N	N	8 weeks of EMDR within 15-week study period
Vonderlin 2025	Cohort study	31.1 (10.6)	76.9	N	N	N	Y	N	12 months
Bartsch 2024	Non-randomised experimental study			N	N	Y	N	N	10 to 12 weeks

**Table 2 brainsci-15-00827-t002:** Summary of clinical research outcomes on BPD in the past 5 years (2019–2025).

Outcome		Global Level of Interest
Mortality	Suicide	●High (ideas/suicidal behaviours, not suicide itself)
	Physical	●Very low
Symptoms	BPD	●Main target of current clinical research
	Comorbidities	●Some comorbidities such as substance and PTSD
Psychosocial functioning		●Low interest (generally as a secondary outcome)
Societal aspects		●Very low

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
