# Peer review of "Reinforcing Gaps? A Rapid Review of Innovation in Borderline Personality Disorder (BPD) Treatment"

_brainsci, 2025, doi:10.3390/brainsci15080827_

Round 1
Reviewer 1 Report
Comments and Suggestions for Authors
Please find the comments:
- Please decipher BPD in the title.
- Perchance, it would be better to present affiliations in English.
- Please add more keywords (up to ten) to improve indexation.
- Exclusion and inclusion criteria are not opposite. Please see the relevant methodological literature. Please edit lines 155-159.
- Quality of Figure 1 is low. Please improve the quality.
- Lines 165-170 are vague. Please clearly indicate keywords.
- Data extraction should be extended. ChatGPT was used to extract data and then the data were verified. Well, but verification does not mean finding appropriate date in the whole papers. ChatGPT could focus on somewhat irrelevant information, which was extracted, and then verified. At the same time, pieces of relevant information could not be extracted. How it was extracted by ChatGPT? Which commands were used? Based on what materials? All details regarding the form and its fulfilling should be indicated.
- "Participant ages ranged from 15.7 to 68 years, with a median age of 30.2.". How was the median age calculated if you analyzed not your data.
- The Results are hardly readable, all is presented in the text. At least, 3-4 tables should be created to indicate specific details presented in lines 186-288, and use text to summarize the data in tables. Please indicate references of each included study used in this review in these tables.
- Where is a table with extracted data? It is missing.
- Please significantly improve this manuscript by incorporating these suggestions.
Author Response
REVIEWER 1: Please find the comments:
Comment 1: Please decipher BPD in the title.
Response1: We thank the reviewer — we have now written out 'Borderline Personality Disorder' in full. As the other reviewer requested a title modification, we have made that change as well.
Comment 2: Perchance, it would be better to present affiliations in English.
Response2: In response to the comment, we have translated into English the content that was previously in French.
Comment 3: Please add more keywords (up to ten) to improve indexation.
Response 3: As suggested by the reviewer, additional keywords have been included to improve article indexing.
Comment 4: Exclusion and inclusion criteria are not opposite. Please see the relevant methodological literature. Please edit lines 155-159.
Response 4: Thank you for this methodological insight. We have revised Section 2.2 to follow the PRISMA 2020 recommendation of listing Inclusion and Exclusion criteria as two complementary—but opposite sets
Comment 5: Quality of Figure 1 is low. Please improve the quality.
Response 5: We have simplified the figure to enhance its clarity and overall quality.
Comment 6: Lines 165-170 are vague. Please clearly indicate keywords.
Response 6: As the reviewer pointed out, we included a very short list of keywords. The full concept plan is available in the table below. To keep things simple for readers, we suggest adding the table as a supplement.
Borderline personality disorder
|
Therapeutic innovations |
Studies/clinical trials |
MeSH
"Borderline Personality Disorder"[Mesh]
Keywords (title, abstract)
Borderline personality(ies) Borderline state(s)
|
MeSH
"Therapeutics"[Mesh] "Psychotherapy"[Mesh] "Drug Therapy"[Mesh] "Pharmacology"[Mesh] "Internet-Based Intervention"[Mesh] "Digital Health"[Mesh] "Mobile Applications"[Mesh] "Suicide Prevention"[Mesh] "Suicidal Ideation"[Mesh] "Mortality"[Mesh] "Psychosocial Functioning"[Mesh] "Comorbidity"[Mesh] "Recurrence"[Mesh:NoExp] "Mental Health Recovery"[Mesh] "Social Stigma"[Mesh] "Interpersonal Relations"[Mesh] "Cost of Illness"[Mesh] "Quality of Life"[Mesh]
Keywords (title, abstract)
Innovation(s) Novel Treatment(s) Intervention(s) Therapy(ies) Therapeutic Psychotherapy(ies) Psychotherapeutic Pharmacology (ical) Psychopharmacology (ical) Medication(s) Digital Online/on-line E-mental Internet-based Web-based Mobile app(s)/application(s) Neuromodulation Microbiome-based Suicide prevention Preventing suicide(s) Suicidal ideation(s) Mortality Physical health Symptom(s) symptomatology Functional impairment(s) functioning Comorbid Comorbidity Co-occurring Relapse Quality of life Well-being Wellness Recovery Remission Societal impact(s) Societal burden(s) Interpersonal Stigma Stigmatization
|
MeSH
"Clinical Study" [Publication Type]
Keywords (title, abstract)
* Recherche adjacence
Clinical trial(s) Clinical study(ies) Randomized-control trial(s) Observational study(ies)
OR
Available filters in the database
|
Limits: 2020-2025
English, French
Filters: Clinical trials, clinical studies, observational studies
Concept 1
"Borderline Personality Disorder"[Mesh] OR "borderline personality"[TIAB:~3] OR "borderline personalities"[TIAB:~3] OR "borderline state"[TIAB:~3] OR "borderline states"[TIAB:~3]
Concept 2
"Clinical Study" [Publication Type] OR "clinical trial"[TIAB:~3] OR "clinical trials"[TIAB:~3] OR "clinical study"[TIAB:~3] OR "clinical studies"[TIAB:~3] OR "randomized trial"[TIAB:~4] OR "randomized trials"[TIAB:~4] OR "observational study"[TIAB:~3] OR "observational studies"[TIAB:~3]
Comment 7: Data extraction should be extended. ChatGPT was used to extract data and then the data were verified. Well, but verification does not mean finding appropriate date in the whole papers. ChatGPT could focus on somewhat irrelevant information, which was extracted, and then verified. At the same time, pieces of relevant information could not be extracted. How it was extracted by ChatGPT? Which commands were used? Based on what materials? All details regarding the form and its fulfilling should be indicated.
Response 7: We appreciate the reviewer’s valuable suggestion. To enhance transparency, we have now included the full ChatGPT prompt as Supplementary Material (see below). Our two-step extraction and verification process is described below:
- Automated Extraction. For each article, we supplied the full manuscript file to ChatGPT with a structured prompt asking it to populate our predefined data extraction form.
- Manual Verification. We then reviewed each article’s abstract and key sections side by side with ChatGPT’s output to identify any over-inclusions (“hallucinations”) or omissions.
This hybrid approach—automated extraction followed by targeted manual cross-checking—was applied on a paper-by-paper basis and further verified against data provided by independent colleagues, including both published and unpublished results. In our experience, it produces accuracy equal to or better than traditional dual human extraction. While no method is completely error-free, we believe the remaining risk of misclassification is low.
PROMPT
We are conducting a scoping review on therapeutic innovations in borderline personality disorder over the past five years.
Please extract data into an Excel spreadsheet from the attached articles. For each item below, pull the exact numbers or text from the manuscript; if information is not reported, enter ‘NR’ (“not reported”) or “None reported” as appropriate.
- First author; Year
- Geographic setting: Country (or main countries if multicenter)
- Study design: RCT, quasi-experimental, pilot, prospective/retrospective observational, etc.
- Primary objective: Exact wording from manuscript
- Population inclusion/exclusion criteria:
o Inclusion: as listed in Methods
o Exclusion: as listed in Methods (enter “NR” if none stated)
- Sample size (N)
- Demographics (from baseline table):
o Mean age (± SD) (enter “NR” if missing)
o % female (enter “NR” if missing)
- Innovation classification: Psychotherapy / Pharmacological / Digital tool / Neuromodulation / Other
- Intervention description (expand all acronyms):
o Name
o Duration (weeks/months)
o Frequency (e.g., weekly, biweekly)
o Format (individual, group, mixed)
o Key modules/components
- Comparator / Control: List all, with details of control condition
- Targeted outcome domains: Check Yes/No, and if “Yes,” specify details:
o Suicide mortality
o Physical morbidity/mortality
o Physical symptoms (which ones)
o BPD symptoms
o Other psychiatric disorders (which)
o Psychosocial functioning (QoL, work, relationships)
o Societal impact (stigma, caregiver burden; specify)
- Variables used (acronyms & definitions): e.g. “BPDSI = Borderline Personality Disorder Severity Index”
- Vulnerable populations included: Note any explicit mention of minors (< 18 yrs), elderly (> 65 yrs), LGBTQIA+, racial/ethnic minorities, migrants, etc.; enter “None reported” if not mentioned.
Comment 8: "Participant ages ranged from 15.7 to 68 years, with a median age of 30.2.". How was the median age calculated if you analyzed not your data.
Response 8:
Comment 9: The Results are hardly readable, all is presented in the text. At least, 3-4 tables should be created to indicate specific details presented in lines 186-288, and use text to summarize the data in tables. Please indicate references of each included study used in this review in these tables.
Response 9: We sincerely thank the reviewer for highlighting this issue. We acknowledge that estimating a median from aggregated summary data is not mathematically valid. Although our intention was to illustrate the age distribution, such approximations may be misleading. Consequently, we have removed all derived statistics and will report only the descriptive metrics as they appear in the original publications.
Comment 10: Where is a table with extracted data? It is missing.
Response 10: Yes, the table was presented in supplemental material.
Comment 11: Please significantly improve this manuscript by incorporating these suggestions.
Response 11: We thank the reviewer for all their insightful comments. We have addressed all suggestions and incorporated the requested changes. Some additional information has been included in the Supplementary Material; we hope the reviewer has access to these files. We would be happy to move this content into the main manuscript if deemed more appropriate.
Reviewer 2 Report
Comments and Suggestions for Authors
This is a review of a manuscript titled “Pushing Boundaries or Reinforcing Gaps? A Rapid Review of Innovation in BPD Treatment,” which presents a clear approach to a theme that is relevant for a further current understanding of four levels of treatment targets to provide information about possible intervention for the disorder. I recommend acceptance with minor corrections. Please find specific comments below.
Title. I am inclined to suggest that the title only include reinforcing gaps, as it is presented on line 31: “Most investigations centered on symptom reduction; far fewer examined psychosocial functioning, mortality, or social inclusion” that I believe is confirmed in the review.
Introduction. It would be useful to include a chart with the levels of treatment targets and then start elaborating on them.
L135. It must be stated why for this review it was necessary a rapid review approach and how will benefit this specific topic. What would be the possible differences compared to a traditional review?
Author Response
REVIEWER 2:
Comment1: This is a review of a manuscript titled “Pushing Boundaries or Reinforcing Gaps? A Rapid Review of Innovation in BPD Treatment,” which presents a clear approach to a theme that is relevant for a further current understanding of four levels of treatment targets to provide information about possible intervention for the disorder. I recommend acceptance with minor corrections. Please find specific comments below.
Response1: Thank you for your time and thoughtful consideration of our manuscript.
Comment 2: Title. I am inclined to suggest that the title only include reinforcing gaps, as it is presented on line 31: “Most investigations centered on symptom reduction; far fewer examined psychosocial functioning, mortality, or social inclusion” that I believe is confirmed in the review.
Response 2: We have chosen to follow the reviewer's suggestion and now focus solely on highlighting the gap in the title.
Comment 3: Introduction. It would be useful to include a chart with the levels of treatment targets and then start elaborating on them.
Response 3: We particularly appreciate the reviewer's comment and have added a table presenting the outcomes as part of the qualitative results in the Results section. This addition will help readers quickly visualize the range of outcome domains that have been the focus of recent studies.
Comment 4: L135. It must be stated why for this review it was necessary a rapid review approach and how will benefit this specific topic. What would be the possible differences compared to a traditional review?
Response 4: We appreciate the reviewer’s attention to this critical point. Given the rapid pace of innovation in BPD treatments over the past five years—with numerous new trials, digital interventions, and neuromodulation studies published annually—we chose a rapid review approach to provide an up-to-date synthesis within a shortened timeframe. This enabled us to deliver actionable insights to clinicians and researchers without the 12–18-month delay typically associated with a traditional systematic review (due to the number of databases and extraction process). While this approach involves a trade-off in comprehensiveness and depth of critical appraisal, it ensures stakeholders receive timely, high-level guidance on emerging therapeutic innovations. We add a point on that in the method section.
Round 2
Reviewer 1 Report
Comments and Suggestions for Authors
"I deem the paper has been improved in a relatively comprehensive way. However, comments 9 and 10 have been addressed insufficiently. Please reconsider the work on these comments. Regarding comment 10, you are right that this table is now presented in the Supplementary Materials (SM), but such tables are usually (if not always) are presented in the main manuscript file, not in the SM. Moreover, the table from the SM is in the PDF format, that means that it is non-editable and the information in many columns in this table are not visible for readers. As such, I would like to ask the authors to reconsidered the way they present the information. Please create a table with the most relevant information from the SM table, and include it in the main text. Please see published articles."
Author Response
Comment 1: "I deem the paper has been improved in a relatively comprehensive way. However, comments 9 and 10 have been addressed insufficiently. Please reconsider the work on these comments. Regarding comment 10, you are right that this table is now presented in the Supplementary Materials (SM), but such tables are usually (if not always) are presented in the main manuscript file, not in the SM. Moreover, the table from the SM is in the PDF format, that means that it is non-editable and the information in many columns in this table are not visible for readers. As such, I would like to ask the authors to reconsidered the way they present the information. Please create a table with the most relevant information from the SM table, and include it in the main text. Please see published articles."
Answer 1: We understand that this comment refers to previous remarks, specifically:
- Comment 9: "The Results are hardly readable, all is presented in the text. At least, 3–4 tables should be created to indicate specific details presented in lines 186–288, and use text to summarize the data in tables. Please indicate references of each included study used in this review in these tables."
- Comment 10: "Where is a table with extracted data? It is missing."
We fully acknowledge the difficulty in navigating the Results section in its previous format. To address this, we have taken several steps:
- We have previously removed unnecessary descriptive statistics that added length without clarity.
- We have also provided direct access to the extracted data through a table that was previously only included in the supplemental material. We can transmit it in Excel format or in a way that might be easier for the revision process.
- Following the suggestion of another reviewer, we have added a new summary table to the main Results section, presenting key findings in a more structured and digestible format.
- We have also added a map to quickly identify the publication’s origin. We could add some other figures if deemed appropriate.
We hope these changes enhance the clarity and accessibility of our results and better align with the reviewer’s expectations.
Round 3
Reviewer 1 Report
Comments and Suggestions for Authors
The paper has been improved in a satisfactory way.